# Phospholipid-Membrane-Based Nanovesicles Acting as Vaccines for Tumor Immunotherapy: Classification, Mechanisms and Applications

**DOI:** 10.3390/pharmaceutics14112446

**Published:** 2022-11-11

**Authors:** Wenjuan Chen, Yali Wu, Jingjing Deng, Zimo Yang, Jiangbin Chen, Qi Tan, Mengfei Guo, Yang Jin

**Affiliations:** 1Department of Respiratory and Critical Care Medicine, Hubei Province Clinical Research Center for Major Respiratory Diseases, NHC Key Laboratory of Pulmonary Diseases, Wuhan Union Hospital, Tongji Medical College, Huazhong University of Science and Technology, 1277 Jiefang Avenue, Wuhan 430022, China; 2Hubei Province Engineering Research Center for Tumor-Targeted Biochemotherapy, MOE Key Laboratory of Biological Targeted Therapy, Wuhan Union Hospital, Tongji Medical College, Huazhong University of Science and Technology, 1277 Jiefang Avenue, Wuhan 430022, China

**Keywords:** liposomes, bacterial membrane vesicles, tumor-derived extracellular vesicles, dendritic-cell-derived extracellular vesicles, vaccine, tumor immunotherapy

## Abstract

Membrane vesicles, a group of nano- or microsized vesicles, can be internalized or interact with the recipient cells, depending on their parental cells, size, structure and content. Membrane vesicles fuse with the target cell membrane, or they bind to the receptors on the cell surface, to transfer special effects. Based on versatile features, they can modulate the functions of immune cells and therefore influence immune responses. In the field of tumor therapeutic applications, phospholipid-membrane-based nanovesicles attract increased interest. Academic institutions and industrial companies are putting in effort to design, modify and apply membrane vesicles as potential tumor vaccines contributing to tumor immunotherapy. This review focuses on the currently most-used types of membrane vesicles (including liposomes, bacterial membrane vesicles, tumor- and dendritic-cell-derived extracellular vesicles) acting as tumor vaccines, and describes the classification, mechanism and application of these nanovesicles.

## 1. Introduction

Nanovesicles composed of lipid bilayers have aroused considerable interest and attention for fundamental study and practical applications. There are two types of phospholipid-membrane-based nanovesicles: pure lipid and/or protein vesicles and comparatively complex cell-membrane-derived vesicles (also called extracellular vesicles (EVs)) [1]. Natural or synthetic lipid and/or protein ingredients make nanovesicles, involving liposomes or proteoliposomes, an ideal model of the membrane system with the advantages of an easy and low-cost production [2]. Furthermore, cell-derived membrane vesicles are regarded as nano- to micrometer-sized containers comprising components such as cellular proteins, nucleic acids and lipids, for the reason that cell plasma or cytosol membranes can enclose these contents while membrane vesicles are secreted [3].

Increasing studies have noticed that EVs are versatile communication tools to establish a link between tumor and host, contributing to tumor development, progression and metastasis. Both tumor and nontumor cells secrete lots of EVs, having a local impact on tumor cells, or traveling through blood vessels to bring about distant influence [4]. Furthermore, bacterial membrane vesicles (BMVs), a type of cell-membrane-based EVs, which are derived from bacteria membrane architecture, own nanoscale vesicle structures containing biomembrane elements of phospholipids. Additionally, BMVs consist of considerable proteins, for example, original bacterial antigens and pathogen-associated molecular pattern components [5]. Therefore, antigen-presenting cells (APCs) are able to recognize and absorb BMVs, and subsequently signaling pathways in immune cells are activated followed by specific immune responses. The molecular components and biological activity of BMVs or further-modified ones render them possible to be a potent vaccine to treat infectious or noninfectious diseases, including cancer [6].

As membrane vesicles have a general characteristic, they contain a lipid bilayer structure that can package hydrophobic and hydrophilic compounds. Despite the practical condition that they are loaded with compounds from the parental cells, a further modification, transformation and fabrication could be conducted as needed to achieve expected goals, such as an enhanced output level, toxicity reduction, targeting improvement, etc. [7]. So far, how membrane vesicles take a part in tumor vaccines is a hot topic. For instance, membrane vesicles conjugated with antitumor immunomodulators have been studied to prevent and treat tumors. The potential of these membrane vesicles as tumor vaccines will be described and discussed in the following section.

Herein, we review the categorization in existence, mechanisms to date and preclinical and clinical applications so far regarding the use of phospholipid-membrane-based nanovesicles as tumor vaccines in the field of tumor immunotherapy.

## 2. Different Origins of Membrane-Based Nanovesicles Are Likely to Act as Tumor Vaccines

There are two primary types of phospholipid-membrane-based nanoparticles, including liposomes and extracellular vesicles (EVs). As EVs are a complex group of biomembrane-based nanostructures, based on their size, biogenesis and function, diverse classification methods have been used. According to the diameter of EVs, they could be roughly classified into small EVs (~100 nm) or large EVs (~100–1000 nm and/or >1000 nm) [8]. Moreover, in terms of biogenesis, EVs can be precisely categorized into exosomes (30–100 nm) produced from the perinuclear luminal membrane and released through multivesicular fusion with the cell membrane, microvesicles (namely microparticles, 100–1000 nm) generated from the cell membrane budding and apoptotic bodies (500–2000 nm) produced through the protrusion of apoptotic cell membrane by dying cells [3,9,10,11]. On the basis of modification or not, natural EVs and engineered ones are included in the family of EVs. Furthermore, EVs have various origins, for example, generated from prokaryotic or eukaryotic cells, or from cell or plasma membranes, since a number of cells are able to secret EVs and subsequently EVs are filled with original components [1].

We then focus on the use of biomembrane-based nanoparticles as a kind of tumor vaccine in the field of tumor immunotherapy or precaution. The classification, structure and composition of four major types of these nanoparticles are going to be exemplified in the following sections, involving liposomes, BMVs, tumor-derived EVs (TEVs) and dendritic-cell-derived EVs (DEVs) (Figure 1).

### 2.1. Liposomes

From a biological point of view, since a cornucopia of lipids and proteins are combined together to naturally form cell membranes, in vitro natural lipids (for example phospholipids) or synthetic components are able to self-assemble into spherical bilayer nanoparticles, namely liposomes. Depending on the preparation methods, the size of the liposomes varies from each other ranging from small (3–5 nm) to giant vesicles (>1 μm) [12]. 

In terms of the biomedical application of liposomes as a tumor vaccine, it is not surprising that simplified membrane vesicle liposomes alone could not perform effective vaccination in various diseases, since the structure and composition are limited and blood circulation times and stability lack satisfaction when they are applied. Thanks to its versatility, scientists take advantage of the properties of phospholipid membranes including chemical (amphiphilicity) and mechanical (stability, permeability and bending and stretching elasticity) peculiarity [13,14]. Therefore, the modification and functionalization of liposomes are necessary and available. Based on seminal reconstitution protocols, liposomes can achieve characteristics and be smart. For instance, liposomes could be stabilized by covering them with densely packed coats such as biocompatible PEG chains (poly (ethylene glycol)) and a crystalline bacterial cell surface layer [15,16]. In addition, according to the charge and specific membrane structure, reconstructed and engineered liposomes are present to mitigate drug delivery problems (safety, efficiency, internalization and targeting ability) [2,17]. Then, the liposome capacity of the efficiently delivering and suitably releasing cargoes (drugs, antigens, siRNA, etc.) can be improved. Taking some construction means as an example, Lian, Shu et al. designed cationic liposomes containing si-CD47 and si-PD-L1 and modified them with EpCAM (epithelial cell adhesion molecule), so they not only enhanced the immune therapeutic efficacy of liposomes, but also improved the targeting ability owing to the overexpression of EpCAM proteins in tumor cells [18].

### 2.2. Bacterial Membrane Vesicles

BMVs are capable of transporting a variety of molecules (proteins, nucleic acids and toxins), and to some extent, the structure of the vesicles and composition of the cargoes differ between Gram-positive and -negative bacteria [19,20]. Both of them exhibit significant effects on innate and adaptive responses. For example, BMVs have been demonstrated to have protective characters against exogenous pathogens or endogenous mutants and have been applied for vaccine exploration [21,22].

For Gram-negative bacteria-derived BMVs, it is widely regarded that the vesicle structure is formed from three stratified layers of the fluid phospholipid bilayer, peptidoglycan cell wall and phospholipid membrane characteristically carrying lipopolysaccharide (LPS) in turn from inside to outside [23]. As a critical participant in bacterial communication and homeostasis, outer membrane vesicles (OMVs) are naturally generated from Gram-negative bacteria, and thus contain pathogen-associated molecular patterns (PAMPs). Therefore, these vesicles, such as pathogen mimetic adjuvants, possess intrinsic immunostimulatory properties acting as a vaccine. Moreover, since membrane vesicles are capable of drug delivery, their natural composition could be enriched by modifying these vesicles with other immunomodulatory agents [24,25,26].

In the aspect of the structure of Gram-positive bacteria-derived BMVs, depending on their own construction, these BMVs lack the package of the outer membrane, while they are coated with rigid peptidoglycan cell walls from the cytoplasmic membrane layer [27]. Hence, these cytoplasmic membrane-generated BMVs are named CyMVs. Although the biological processes of Gram-positive BMVs are less understood than Gram-negative BMVs, they are still being taken into consideration for vaccine applications. The vaccination efficacy of BMVs from certain bacterial strains has been estimated as a strategy to fight against Gram-positive bacteria, and vaccine safety has been detected as well through monitoring toxin-specific antibodies. For the purpose of controlling and decreasing toxicity, approaches to producing genetically engineered BMVs emerge one after another. For instance, Wang, Xiaogang et al. purified BMVs from Staphylococcus aureus to establish an *S. aureus* vaccine platform which could package cytosolic and secreted proteins, such as cytolysins and phenol-soluble modulins, contributing to BMVs biogenesis and detoxification [28].

### 2.3. Tumor-Cell-Derived EVs

As EVs play a crucial role in cell–cell communication, tumor-cell-derived EVs (TEVs) take part in tumorous progression, tumor microenvironment modulation and distant metastasis, and even accelerate these processes [29,30]. 

Compared with liposomes, the contents and structure of EVs are relatively complicated. Meanwhile, the innate composition of TEVs inherited from parental tumor cells has gained a lot of attention. Considerable evidence from recent studies has demonstrated that there are lots of oncoproteins (such as phosphorylated epidermal growth factor receptor (EGFR), vascular endothelial growth factor (VEGF), stromal-cell-derived factor 1 (SDF1), etc.) and oncogenic RNA (in particular miRNAs) in TEVs, making a contribution to tumor growth, invasion, migration and angiogenesis [31,32,33,34]. For these TEVs carrying parent-cell-specific signatures, they permit an interaction between the targeting cells and are potentially used to be explored or confirmed as biomarkers in liquid biopsies for distinctive diseases [35]. Due to the existing biological qualities of TEVs, development for them acting as potential therapeutic strategies becomes suitable and overwhelming. For example, approaches inhibiting the preparation and secretion of native EVs or loading TEVs with antitumor drugs or RNAs which encode information of disrupting tumor progression are emerging rapidly [36]. 

Since TEVs are assembled and packaged in a tumor-cell-specific manner, the components (in particular, the tumor antigens) are distinct from the EVs from mesenchymal stem cells or blood cells. Peptides in antigen-rich TEVs could be presented by major histocompatibility class (MHC) receptors and then cause the following interaction with immune cells (CD8^+^, CD4^+^ T cells or natural killer cells (NKs)) to help enhance cancer immunotherapy [22,37]. Additionally, to maintain biocompatibility and elevate the immunogenicity of TEVs, the formulation should be well designed, in order not only to guarantee biosecurity, but also to positively regulate antitumor immune responses and deliver therapeutics to tumor sites. In numerous basic or clinical studies, engineered or nonengineered, autologous or nonautologous TEVs have been fabricated and administrated to biomedical model species or human subjects [38].

### 2.4. Dendritic-Cell-Derived EVs

Cell-membrane-based EVs are also secreted by immune cells in tumor tissues to contact the rest of the immune cells and tumor cells [39]. Immune-cell-derived EVs play a crucial role in cell–cell communication for tumor cells to escape from immunological surveillance and for further studies to design potential tumor vaccines [40]. Among these EVs, a relatively large focus has been put on dendritic cells (DCs)-derived EVs (DEVs or dexosomes) as a tumor vaccine, and thus DEVs have become an attractive candidate for tumor immunotherapies.

In a typical immune loop, antigen-presenting cells (APCs), among which DCs are the most potent APCs, prime the immune responses when the immune system defends against tumorous, viral or bacterial diseases [41]. It is commonly believed that DCs expressing MHC class I or II molecules on the cell surface interact with other immune cells such as CD8^+^ or CD4^+^ T lymphocytes and natural killer cells to initiate an immune reaction [42,43]. During EVs secretion from DCs, it has been found that EVs also encapsulate peptide-MHC complexes (p-MHC) together with costimulatory molecules such as CD40 and CD86 [44,45,46]. Additionally, it could carry abundant immunoregulatory cargos, including cytokines, complement factors and immunosuppressive or active molecular mediators [47]. It has been verified that in the absence of APCs, DEVs own the ability to activate T cells since they can directly interact with T-cell receptor (TCR) complexes [48]. Flourishing evidence proves that the capability of modified or engineered DEVs to elicit antigen-specific antitumor immune responses could be enhanced when costimulatory factors are upregulated and immunoregulatory/immunosuppressive signals are reduced [49]. It is no wonder that DEVs continue to be a promising nanomaterial for further research on vaccination in tumor immunotherapy. 

## 3. Membrane Vesicles Work to Bring a Stone to the Building of Tumor Immunotherapy from the Perspective of Basic Mechanisms

The potential role of membrane vesicles as tumor vaccines in tumorous immune responses has been extensively documented and exploited. In most cases, the membrane vesicles listed above have been well designed, engineered and modified, before they are used to treat cancer or other diseases. The function of membrane vesicles acting in immune regulation is likely due to diverse reasons, including their presentation or transfer of antigenic peptides (tumor-specific antigens (TSAs) or tumor-associated antigens (TAAs)), delivery of adjuvants, immunomodulatory molecules or cytosolic DNA, gene-expression manipulation by miRNA or plasmids and induction of favorable signaling pathways by ligands expressed on the surface of the vesicles [11]. Generally speaking, in terms of mechanisms based on immunity, membrane vesicle vaccines induce the activation and maturation of DCs, provoke T cells and arouse immune memory, to play a role in tumor-suppressive efficacy and prolonged survival of multiple tumor biological models. In the following review portion, we discuss the key mechanisms primarily focusing on the change in immune responses under the administration of these four types of potential membrane vesicle vaccines at the level of basic medicine (Figure 2).

### 3.1. The Mechanisms of Liposomes as a Tumor Vaccine

As typical synthetic phospholipid-based vectors, liposomes have outstanding properties, such as flexible characteristics (including excellent elasticity and changeable degrees of liquidity), ideal biocompatibility and low toxicity [50,51]. Although it is tough for simple and hollow lipid-membrane-based nanovesicles to be directly used as a vaccine, the versatile features make it possible for them to achieve functions of tumor vaccines via adventurous artificial liposome modification. 

Currently, increasing studies have exploited or are developing liposomes encapsulated with antigens, antibodies, adjuvants or chemical drugs depending on their hydrophilic and hydrophobic moieties. To conduct feasible applications of liposomes in tumor immunotherapy, cationic liposomes containing synthetic tumor long peptides have been designed and prepared, and such nanoparticle vaccines have been demonstrated to stimulate both CD4^+^ and CD8^+^ T cells and enhance the efficacy of a checkpoint inhibitor for lung cancer [52]. Apart from the combination of liposomes and checkpoint inhibitors, liposomes can carry some inhibitors to improve the therapeutic effects of a checkpoint blockade. For example, Tu, Kun et al. verified that liposomes codelivering BMS-202 (one of the PD-L1 inhibitors) and chidamide (CHI, an epigenetic modulator inducing immunogenic cell death) boosted antitumor immune responses through CHI-related increased immunogenicity and a BMS-202-mediated PD-L1 intercept [53]. In addition to carrying tumor antigens, liposomes could be firstly modified with cholesterol cationic peptide DP7, which makes liposomes successfully transfer mRNA into DCs, and then the production of the DCs antigen presentation activity and secretion of proinflammatory cytokines were found to elevate, further contributing to antigen-specific lymphocyte reactions and tumor inhibition [54]. Additionally, many more complex modifications to improve the antitumor effects of liposomes has also drawn lots of attention. For example, Liang, Ruijing et al. fabricated a gold nanoplatform coated with liposomes and meanwhile tethered aCD11c (a ligand to target DCs), monophosphoryl lipid A (a strong adjuvant to induce immune responses) and peptides from tyrosinase-related protein 2 (a melanoma antigen to stimulate specific immune activity) [55]. The researchers have taken the superiorities of liposomes and endowed them with immunogenic functions, so as to activate a complete antitumor immune response involving DCs maturation, cytotoxic CD8^+^ T lymphocytes (CTLs) activation and other routes of the immune system.

### 3.2. The Immunologic Mechanisms Triggered by Bacterial Membrane Vesicles

Owing to the microbe-associated molecular patterns (MAMPs) and nanoscale structure, BMVs have a high immunogenicity. They are developed into vaccines against infectious or noninfectious diseases through chemical, physical and genetic modifications. Thanks to the evolution of these modifications, BMVs can be endowed with more appropriate and ideal functions or properties to fight against certain diseases, for instance, increased safety and efficacious therapeutic ability.

Speaking of activating the influence of BMVs on immunity, it is essential to mention that vesicle contents, including lipoproteins, nucleic acids and other immunostimulatory MAMPs, may be recognized by the relevant receptors such as pattern recognition receptors (PRR), which are detected to express on the surface of epithelial cells and immune cells (in particular DCs) [56]. The ligand–receptor combination triggers the immune responses, whereas the form and intensity of these responses vary from each other, relying on the composition and structure of the BMVs to some extent [25]. As extensively reported, Toll-like receptor 4 (TLR4), one of the PRR, has been verified to interact with BMVs containing MAMPs such as LPS via the induction of PRR signaling, for example TLR4-dependent CXCL8 (CXC-chemokine ligand 8, also known as IL-8) production, MAPK (mitogen-activated protein kinase) -AP1 (activator protein 1) signaling, etc. [57,58]. Moreover, when peptidoglycan (PG) existing on the membrane wall of BMVs dominates the ligand–receptor communication, the NOD1 (nucleotide binding oligomerization domain 1), known for an intracellular sensor, could be activated via recognizing PG, and the ascent of the cytokines CXCL2 expression level has been found [59]. 

In addition to the innate immune system of which MAMPs in BMVs can drive the generation, BMVs can effectively incite cellular and mucosal immunity. For instance, engineered BMVs with the expression of IL-10 on the surface and the encapsulation of HPV16 E17 protein into them have been demonstrated to prevent the exhaustion of T cells, presentation of p-MHC I complexes and activation of CD8^+^ T cells transforming into CTLs, causing the inhibition of tumor growth and metastasis and implying a novel tumor immunotherapeutic strategy [60].

Among components of bacterial nanosized vaccine platforms, outer membrane vesicles (OMVs) produced by Gram-negative bacteria are the most overlooked [21]. Numerous studies have employed OMVs as therapeutic agents to treat cancer by means of the induction of antitumor immune responses. In an interesting study, bacterial OMVs were loaded with polymeric micelles (a kind of liposome) which are responsible for chemotherapeutic function and even help achieve the stimulation of T cells, and such modified OMVs elicited stronger immune responses, resulting in an increased survival rate and decreased tumor growth and metastasis in rodents [49]. OMVs can carry abundant quantities of LPS or other immunomodulatory factors, whose capacity for immune activation and modulation has been used to reduce tumor growth [61]. However, LPS is a double-edged sword, because it is a kind of strong inducer of inflammation and thus is the main reason that OMVs have biotoxicity [62]. Therefore, the biosecurity of OMVs should be taken into much consideration. Hopefully, incomplete LPS structures in genetically engineered OMVs cause toxicity to decline and keep the immunogenicity at the same time [63]. In the study conducted by Kim, Oh Youn et al., OMVs, extracted from genetically modified Escherichia coli (E. coli) whose genes encoding the lipid component of LPS were dampened ahead of time, exhibited biosafety, a targeting ability and ideal tumor attenuation after systematical administration, and induced the production of CXCL10 and IFN-γ (interferon-γ) in colon cancer cell-bearing mice [64]. Although the immune-associated mechanisms involved are not completely understood, the potential of OMVs as a classical tool for tumor vaccine design is worth being highlighted.

### 3.3. The Mechanisms of Tumor-Cell-Derived EVs Designed to Be Tumor Vaccines

TEVs have emerged as a booming drug- or compound-delivery system for the development of tumor treatment, including chemotherapy and immunotherapy.

Generally speaking, TEVs suppress innate and/or adaptive antitumor immune responses. In the aspect of innate immunity, natural TEVs have been found to influence macrophage polarization and inhibit the activation and proliferation of NK cells. Bardi, Gina T et al. demonstrated that melanoma exosomes could upregulate the cytokine secretion of the M2 phenotype and thus induce a protumor macrophage activation phenotype mixture [65]. In addition, as Liu, Cunren et al. have studied the impact of carcinoma exosomes on NK cell function, TEVs could suppress NK cell cytotoxic activity by inhibiting the release of perforin and the expression of cyclin D3 [66]. Referring to adaptive immunity, TEVs are loaded with a composition similar to the parental cells and it is well known that tumor cells express the surface-protein-programmed death-ligand 1 (PD-L1) to evade immune detection. TEVs have the same topology of PD-L1 as that on the cell membrane, and TEVs could dumb the function of CD8^+^ T cells to facilitate tumor development [67]. Chen, Zhenzhen et al. demonstrated that TEVs from diffuse large B cell lymphoma were detected to elevate the expression of PD-1 on the surface of T cells, hence acting as an immunosuppressive mediator [68].

Conversely, in certain situations, TEVs could exert a tumor-inhibiting role through the activation of the antitumor immune system. Interestingly, the formation of TEVs may influence their behavior in promoting or suppressing tumor cells. It has been revealed that TEVs derived from radiation-irritated lung cancer cells have antitumor therapeutic effects via tumor microenvironment (TME) remodeling [69]. Furthermore, tumor antigen profiles in TEVs endow them with the potential to be tumor vaccines. TEVs carry damage-associated molecular patterns (DAMPs) such as heat shock proteins (HSP) and subsequently activate the immune cells, specifically DCs [70]. As Ma, Jingwei et al. have reported, TEVs can increase lysosomal pH and drive lysosomal centripetal migration, promoting the formation of p-MHC I complexes by DCs, which present antigen peptides to CD8^+^ T cells [71]. In addition, HSP70-overexpressing tumor cells could release EVs rich in HSP70 proteins, and these EVs become a reservoir of extracellular HSP70 to activate and augment the cytotoxic activity of NK cells, or to play an immunostimulatory role in antigen presentation and T cell priming [72,73]. Not surprisingly, TEVs, especially batch-to-batch engineered or modified vesicles, exhibit a splendid character for tumor antigen delivery. The goals to engineer or modify TEVs are various, including the enhancement of pharmacokinetic and/or pharmacodynamic characteristics, maintenance of biocompatibility, decrease in biotoxicity or improvement of immunogenicity [22]. These tumorous antigens or peptides that TEVs possess can be presented by APCs forming p-MHC compounds to incite interactions with other immune cells. In addition to antigens associated with the activation of DCs, the DNA components in TEVs can also activate DCs in some signaling pathways. In the recent few years, increasing amounts of researchers have revealed the connection between epigenetic mechanisms and immunoregulation within a tumorous niche, aiming to boost antitumor immune responses under the impact of epigenetic therapy [74]. Some strategies have been conducted to genetically modify TEVs or other membrane nanovesicles with epigenetic agents performing gene silencing through various signaling pathways, for example, altered micro RNAs, histones and abnormal DNA methylation [75,76]. In addition, the stimulator of interferon genes (STING) is an important signaling pathway because it plays a vital part in the differentiation and manipulation of myeloid-derived suppressor cells (MDSCs) and EVs generate specific T-cell responses via the aforementioned pathway to perform anticancer effects [77,78]. Y. Kitai et al. treated breast cancer cells with an inhibitor of topoisomerase I before they collected and extracted cancer-derived EVs which could contain a larger volume of DNA, and then DCs were activated to elicit antitumor immunity by the TEVs via the cGAS-STING pathway [79]. Moreover, our previous study concluded that TEVs packaging the metabolism-relevant inhibitor Fluvastatin can suppress cancer cells via reversing immunosuppressive TME by increasing the infiltration of CTLs, M1-macrophages and activated NK cells and downregulating immunosuppressive cells including M2-macrophages, Tregs and MDSCs [80]. Therefore, not spontaneous TEVs, but these processed ones have the potential to become an ideal tumor vaccine.

### 3.4. The Potential Mechanisms of Dendritic-Cell-Derived EVs Acting as a Tumor Vaccine

As described ahead, cell-free vectors DEVs also show therapeutic promise, since they directly present the specific antigen to T cells and the activation of these effector cells is stimulated. DEVs bear peptide-MHC complexes and inherit the properties that parental DCs have. A number of fundamental and clinical studies load tumor antigen peptides to DEVs, in a bid to suppress tumor growth by evoking helpful antitumor immunity. Zuo, Bingfeng et al. painted DEVs with hepatocellular carcinoma peptides, α-fetoprotein epitope and an immunoadjuvant (which was nucleosome-binding protein 1) to promote the recruitment and activation of DCs in tumor sites, achieving the cross-presentation of tumor neoantigens and causing tumor eradication [81]. Except containing peptides of a signal tumor, DEVs could be engineered to carry tumor antigens from multiple tumors. From the perspective of Tian, Xin and his colleagues, DEVs bearing dual antigens from melanoma and lung adenocarcinoma were generated to induce specific tumor-rejected immune responses owing to the upregulation and stimulation of CTLs [82]. Alternatively, antigens in DEVs could be ingested by other APCs, and then APCs deliver such antigenic EVs to the specific sites to elicit the priming of T cell and/or B cell crosstalk [83]. In some in vitro studies, it has been shown that DEVs have to be recaptured by DCs and then transfer antigen information so as to accomplish the interaction with T cells [84].

Despite the fact that the type of antigens carried by DEVs is crucial for them to specifically initiate an immune response, the size of EVs should also be considered. Wahlund, Casper J E et al. unraveled the divergence of DC-derived exosomes and microvesicles. They found that exosomes carried much more ovalbumin (OVA), whereas the OVA in microvesicles was hardly detectable [85]. Hence DC-derived exosomes show a stronger capacity to induce antigen-specific T cell responses or antibody production, indicating that exosomes own much more obvious immunogenicity, and more importantly implying that subpopulations of EVs need be carefully concerned before design, manipulation and application. In addition to antigen peptides and the size of DEVs, the origin of DEVs is also important, for the reason that there are diverse developmental stages during the maturation of DCs. Admyre, Charlotte et al. evaluated the T cell stimulatory capacity with exosomes from lipopolysaccharide-matured DCs in advance and found an enhanced level of the activated function of T cells compared with exosomes from immature DCs [86].

## 4. The Current Advances of Representative Membrane Vesicles as Tumor Vaccines in Preclinical Studies

Owing to the unceasing and evolving processes of understanding how immune cells interact with each other and communicate with tumor cells or other cells, continuously developing membrane vesicles are emerging as tumor vaccines and are utilized for immunotherapy. Phospholipid-membrane-based nanovesicles can be loaded with antigenic payloads, immunomodulatory factors or therapeutic drugs, or receive epigenetically/genetically engineered reprogramming to overexpress or downregulate the expression of certain molecules for the antitumor purpose. These nanomaterials have curative and tumor-killing effects in different types of tumors through stimulating antitumor immune responses and reversing the immunosuppressive microenvironment. Additionally, to generate an effective, qualified or even excellent tumor vaccine, it is critical to evoke the specific, robust and long-term immunity and induce the attenuating of immunosuppressive factors. Here, an overview of the multiple vaccines from modified and/or engineered membrane vesicles against tumors in the recent decade is provided. 

### 4.1. Synthetic Liposomes Used as a Tumor Vaccine

Through the versatile and operational characteristics of liposomes, there is no doubt that liposomes are widely applied in antitumor treatment. Existing material design strategies for liposomes acting as tumor vaccines are generally to functionalize them with immunogenicity, targeting ability and immunomodulatory capacity by carrying adjuvants, tumor-specific antigens/peptides, molecules specifically recognizing certain receptors or different RNA types (Table 1). In addition, during fabrication processes, emphasis should be put on internalization efficiency, circulation traffic, biocompatibility and security. We list some representative studies about liposomes used for tumor immunotherapy in recent years (Table 1). The design approaches, application in tumors and immune-associated mechanisms are briefly described in Table 1.

### 4.2. The Design and Engineering Modification of Bacterial Membrane Vesicles

BMVs are designed not only to target tumor sites and provoke effective immune reactions, but also to reduce inflammatory toxicity and specifically kill tumor cells. Comprehensive studies revealing various fabrications of BMVs as tumor vaccines are documented in Table 2.

BMVs have been reported to induce the production of antitumor-associated cytokines such as IFN-γ, TNF-α and IL-12 [25,101]. Except the immunological functions of BMVs, BMVs have a rigid membrane structure which makes nanoplatforms stabilized. Moreover, it is easy to produce BMVs by fermentation and purification procedures, and it is workable to acquire genetically engineered BMVs [102]. Owing to theses intrinsic properties, it is not surprising that integrating BMVs into tumor vaccines is appealing. BMVs can be originated from various types of bacteria, while nanosized OMVs secreted by Gram-negative bacteria, especially E. coli, are mostly developed to be a tumor vaccine. Before collecting BMVs, original bacteria can be genetically engineered by being transfected with RNA interference, miRNAs, etc., for some purpose, such as detoxification. Additionally, a modification is commonly applied after BMVs are ultracentrifuged and purified, for example, binding them with tumor antigens or immunoadjuvants (Table 2). 

**Table 2 pharmaceutics-14-02446-t002:** Fundamental applications of BMVs as tumor vaccines.

Parent Bacteria	Modification Strategy	Targeting Tumor Types	Mechanisms and Outcomes	Year, Reference
*Escherichia coli (E. coli)*	Bacteria and liposome biohybrid vaccine combined with tumor antigen and adjuvant	Colorectal cancer	Increased expression of CD40, CD80 and CD86 on BMDCs and enhanced infiltration of CD8^+^ T cell	2021, [103]
*E. coli*	Genetically engineered OMVs binding with L7Ae (RNA binding protein) and listeriolysin O (lysosomal escape protein)	Melanoma; colon cancer	Listeriolysin O-mediated endosomal escape contributes to cross-presentation of DCs; induction of a long-term immune memory.	2022, [104]
*E. coli*	OMVs fused with thylakoid membranes from spinach	Colon cancer; breast cancer	Photodynamic effects from thylakoid cause tumor destruction, resulting in release of TAAs and DAMPs presented by DCs and inducing tumor-specific CD8^+^ T cell responses.	2022, [105]
*E. coli*	OMVs fused with protein cytolysin A	Pulmonary metastatic melanoma; colon cancer	The antigen-bearing OMVs stimulate DCs maturation and protect animals against tumorous rechallenge.	2022, [106]
*E. coli*	Conjunctive products of OMVs, Mal and 1-MT (IDO inhibitor)	Colon cancer	The nanoparticles bind to tumor antigens and overcome the immune inhibition of IDO on effector T cells.	2022, [107]
*E. coli*	OMVs fused with ClyA protein and decorated with tag/catcher protein pairs	Lung melanoma metastasis; colorectal cancer	The vaccine platform “Plug-and-Display” technology displays the tumor antigens and induces innate and specific T-cell-mediated immune responses.	2021, [108]
*E. coli*	Synthetic OMVs combined with TEVs	Melanoma	Synthetic OMVs have barely any systemic proinflammatory responses; The combined membrane vesicles activate BMDCs, Th-1 T cells and balance antibody production; efficacy of antiPD-1 inhibitor is improved.	2021, [61]
*Salmonella Typhimurium*	A eukaryotic–prokaryotic vesicle (EPV) nanoplatform containing TEVs and OMVs	Melanoma	It is verified to be a prevention vaccine to trigger antitumor memory immune responses; photothermal effects are motivated by combination with EPV through DCs maturation and production of TNF-α and IL-12.	2020, [109]
*E. coli*	OMVs modified by insertion of the ectodomain of PD1	Colon cancer; melanoma	OMVs bind to PD-L1 on the tumor cell surface and thus protect T cells from PD1/PD-L1 axis; OMVs induce the accumulation of effector T cells in TME.	2020, [110]
*Salmonella Typhimurium*	OMVs from *Salmonella Typhimurium*	Colorectal carcinoma; hepatocellular carcinoma; breast cancer	OMVs enhance recruitment of NK cells through upregulation of caspase-3, Beclin-1 and CD49b.	2021, [111]

Abbreviations: Mal, maleimide; 1-MT, 1-methyl-tryptophan; IDO, indoleamine 2, 3-dioxygenase.

### 4.3. The Proper and Multiple Strategies for Producing Tumor Vaccines from Tumor-Cell-Derived EVs

As TEVs play a crucial biological role in tumorous biogenesis, development and metastasis, it cannot be denied that antitumor strategies vary from each other depending on the distinctive mechanisms, for example, suppressing TEV secretion, interrupting TEV uptake by recipient cells or delivering functional cargoes [4]. Apart from these approaches, the applications of TEVs as tumor vaccines in the last few years are summarized below (Table 3). 

Notably, compared with the composition of liposomes or BMVs, TEVs innately contain a plethora of bioactive molecules, especially tumor antigens (TAAs and TSAs), which are critical when TEVs are involved in immunity modulation. Nevertheless, naturally autosecreted TEVs cannot be directly applied in tumor immunotherapy, since they have an important part in tumor biogenesis and progress as we have mentioned above. Therefore, engineering or modification methods are essential to fully obtain their advantages and potentials as a tumor vaccine. Before isolation is conducted, tumor-cell-related genetic engineering can be performed, including gene/protein-encoding interference (miRNAs, mRNA, siRNAs and others) and genome editing (plasmid DNAs) [1]. Furthermore, therapeutic drugs, adjuvants or antigen peptides can also be attached to TEVs before or after isolation so as to improve their capacity to be a feasible tumor vaccine.

### 4.4. Transforming Dendritic-Cell-Derived EVs into an Ideal Tumor Vaccine

Since DEVs partially similar to DCs participate in the initiation of proper antitumor immune responses of T cells consisting of antigen recognition (p-MHC complexes-TCR communication performs an important role), costimulation and/or production of T-cell stimulatory cytokines, it is inevitable to further consider the DEVs used in tumor immunotherapy [22]. Additionally, DEVs are able to trigger or modulate innate antitumor immune responses, for example, the activation of NK cells [123]. Rather than the direct use of naked EVs generated from different types of DCs in tumor immunotherapy, DEVs also undergo multiple design means. Similar to the approaches described in the previous section, DEVs can be loaded with therapeutic sensitizers, immunomodulators, adjuvants, tumor-specific peptides or small molecules targeting some signaling pathway. It is noted that not only do TEVs carry tumor-associated antigens (TAAs), but DEVs can also be pulsed with TAAs when they are generated from patient tissues [4].

We document representative preclinical studies in recent years about DEVs modified to be a tumor vaccine and give a brief description of immune-related mechanisms and how DEVs evoke potent antitumor immunity (Table 4).

## 5. The Clinical Applications Relevant to Tumor Vaccines of Biomembrane-Based Nanovesicles Are Developing

The aforementioned applications of membrane vesicles as potential tumor vaccines are mainly focused on the fundamental level, that is to say, applied in vitro or in animal models. Due to the preponderance of drug/molecule-delivery efficiency, internalization ability, ease of modifying and biocompatibility, these membrane-based nanovesicles (liposomes, BMVs, TEVs and DEVs) exhibit potential for clinical and biomedical translation. For the purpose of the clinical evaluation of tumor vaccine efficacy, tumor status (tumor growth, invasion and metastasis), survival indexes (overall survival and progression-free survival) and systemic and local immunity (implying how vaccines work and influence the human body) are involved in the assessment system. In addition to therapeutic effect appraisals, adverse effects caused by vaccines are always crucial and need careful observation [131]. Since membrane vesicles act as vaccines potentially influencing patient immunity, monitoring immune responses is essential for the sake of security during the clinical trials, not just for the evaluation of curative effects. On account of the fact that a wide range of clinical trials has been undertaken, the advances and meanings of complete or developing clinical studies to date are going to be reviewed in the following section. Additionally, the brief synopsis of these representative clinical trials appearing in the context can be seen in Table 5.

Liposomes, amongst membrane vesicles having a relatively long history, are used for tumor vaccines in a number of clinical trials. In the Netherlands, eligible ovarian cancer patients are treated with a liposome-formulated mRNA vaccine that can encode TAAs and later peripheral blood mononuclear cells, and the intratumoral accumulation of immune cells are used to determine immune responses (NCT04163094). Additionally, liposome-based mRNA vaccines exhibit fewer side effects, a higher stability and stronger efficacy than the sole mRNA vaccines in cancer treatment, including lung cancer and colon cancer (NCT01915524) [132,133]. Furthermore, a liposomal vaccine called Lipovaxin-MM in a completed phase I trial has been demonstrated to be safe and effective in metastatic melanoma participants (NCT01052142). Despite the potentially effective vaccines of simple-ingredient liposomal systems, liposomes can be designed to be encapsulated with several components performing their special duties. In a phase I trial, a complex nanovesicle vaccine named DPX-0907 based on IVT’s DepoVax™ (DPX) formulation was applied to ovarian, breast and prostatic cancers, which consisted of tumor and T helper peptides, a polynucleotide adjuvant and lipid components, respectively (NCT01095848). Although the safety was confirmed in the phase I study, a subsequent study to determine the treatment effects need to be carefully planned and the results remain unknown.

There have been a batch of clinical studies that used bacteria or modified ones as potent tumor vaccines to inherently modulate and stimulate antitumor immune responses or reverse immunosuppressive immunity. In spite of some clinical studies using bacteria for tumor vaccines (NCT00623831, NCT02010203, NCT03762291, etc.), bacteria may pose a risk for infection during the treatment, so applying bacteria-derived vesicles that contain concise components seems to be safer [134]. Bacteria contain lots of unnecessary or even toxic components, so it is welcomed through discarding the “dross” and meanwhile selecting the “essence”. Despite the current condition that lots of clinical trials about BMVs emerge to cure infectious diseases, the existing registered clinical trials about BMVs acting as a tumor vaccine are limited, indicating that there are many challenges setting obstacles for tumor vaccines in clinical translation [135]. Meanwhile, lots of concerns have been raised about the biosafety of BMVs. For example, the introduction of LPS or other proinflammatory factors in BMVs may potentially cause an inflammatory storm. Therefore, a variety of methods have been introduced to diminish the component toxicity of BMVs to decrease the incidence of systemic inflammatory responses [136,137].

The parental materials for TEV fabrication applied in patients are various, for example using autologous tumor cells or synthesizing tumor neoantigens loaded in TEVs. In 2019, we published our clinical data about autologous tumor-cell-derived microparticles encapsulated with methotrexate (aTMPs-MTX) used in advanced lung cancer patients with malignant pleural effusion [138]. The clinical results showed that aTMPs-MTX not only decreased the pleural effusion volume but also activated patient antitumor immunity (for example, the upregulation of CTLs) (NCT02657460). Furthermore, TEVs can carry other bioactive byproducts to play a role in the modulation of antitumor immunity in clinical studies. For instance, in a phase I clinical trial, exosomes from patient glioma tissues encapsulated in small diffusion chambers can gently release tumor antigens and perform the activation of immune cells (NCT01550523). Despite the fact that some synthetic TEVs as tumor vaccines seem to be effective in prolonging survival, combination therapy in clinical trials is not unusual. In a phase I trial (NCT00020462), patients with follicular lymphoma receive the therapy strategy combining the autologous tumor cell vaccine and IL-2. Additionally, hybrid or chimeric EVs vaccines exist in clinical studies. Currently, an early phase 1 trial is being conducted, referring to the membrane vesicle combination of chimeric exosomal tumor vaccines prepared from bladder tumor tissues and peripheral blood immune cells (DCs or macrophages) (NCT05559177). However, the treatment effects of incomplete trials remain to be detected.

A decade ago, the FDA approved monocyte-derived DCs pulsed with the antigen prostatic acid phosphatase and the immunomodulatory factor GM-CSF as a form of cellular immunotherapy against prostate cancer, which was demonstrated to be effective in phase III trials (NCT00065442) [139]. In consideration of the functions of DCs being potent tumor vaccines and the novel promising cell-free EVs from DCs, Narita, Miwako et al. performed and reported phase I/II clinical trials (NCT02693236) for patients diagnosed with esophageal cancer, in which DEVs were isolated from antigen peptide-pulsed DCs, and then CTLs were found activated later after the administration of DEVs in patients, while the efficacy of DEVs remained unknown [140]. Another phase II clinical trial in France focused on the immunotherapeutic effects of metronomic cyclophosphamide followed by tumor antigen-loaded DEVs, whose safety and feasibility on unresectable nonsmall cell lung cancer was verified in a phase I trial (NCT01159288). It is notable that not all the trials showed the desired efficacy, which was mainly attributed to the low immune stimulatory capacities of these synthetic membrane vesicles. To relieve or solve the obstacles occurring in the clinical application of membrane vesicle vaccines, the design of the vaccine itself, timing and frequency of vaccination, routes and sites of injection and vaccine doses should be gradually and rigorously established.

## 6. Outlooks

Phospholipid-membrane-based vesicles own evident features making them attractive for basic or clinical applications. With the membrane properties, membrane vesicles are born to be flexible and easy-to-carry cargoes or internalized by recipient cells. As a result, they have inevitably become a promising drug/molecule-delivery platform. Nevertheless, due to the different sources of membrane vesicles, the function and capacities are distinctive from each other, so researchers should properly design strategies according to the actual situations so as to take full advantage of nanosized tumor vaccines, rather than applying them mechanically. There are several approaches concluded as follows to improve the therapeutic successfulness of tumor vaccines: 1. standardized preparation techniques in order to improve the uniformity of nanovesicles including their size and contents; 2. detoxicating toxic components and purifying vesicles to wipe off impurities; 3. selecting a specific and conducive cargo, including antigen peptides (such as hot-discussed personalized neoantigens), immunomodulatory factors, chemical drugs and nucleic acids (such as mRNA, miRNA and siRNA encoding with specific functions); 4. The appropriate modification of vesicle surface or intercellular compartments. It is believed that the advances of membrane vesicles used in tumor immunotherapy are endless.

Although a number of preclinical studies contribute to exploiting developing methods in order to design membrane vesicles as ideal tumor vaccines, several reasons as follows may set obstacles to the clinical translation of membrane vesicles. 1. in the aspect of biosafety, complex contents in vesicles may not only cause side effects (for example, impaired glucose tolerance and fasting hyperglycemia) but may also be difficult to metabolize, thus prolonging in vivo residence, indicating that the toxicology and pharmacokinetics should be further carefully concerned during the intervention [51,141,142]; 2. despite ongoing efforts to improve the procedures associated with the isolation and storage of membrane nanovesicles, the homogeneity of extraction is hard to ensure and the quality control still remains problematic [143]; 3. the technique causes the therapeutic membrane vesicles to be expensive and such curative approaches may raise the cost burden to patients [22]; 4. due to the physiological uniqueness of each patient and/or low immunogenicity of nanoscale tumor vaccines, the precision medicine and/or therapeutic effects are difficult to be determined and guaranteed; 5. ethical issues may hinder the clinical application, for the reason of the potential adverse impact of genetically modified membrane vesicles, such as uncomfortable feelings (fatigue, fever, etc.) or even continuous tumor development probably caused by the cargoes in the membrane nanovesicles [131,144]; 6. the retention in the systemic circulation is not durable and nanovesicles may be rapidly cleared by the mononuclear phagocytic system, resulting in a low efficiency and high frequency of vaccination, so novel therapeutic systems such as nanovesicles camouflaged with red blood cell or platelet membranes are being studied [141,145]. Taken together, uniform and definite criteria are difficult to accomplish, and researchers have to be faced with and overcome existing hindrances. In summary, although there is still an arduous and long road to go for the development of membrane-vesicle-based tumor vaccines, it does not keep membrane vesicles from being promising tumor vaccines.

## Figures and Tables

**Figure 1 pharmaceutics-14-02446-f001:**
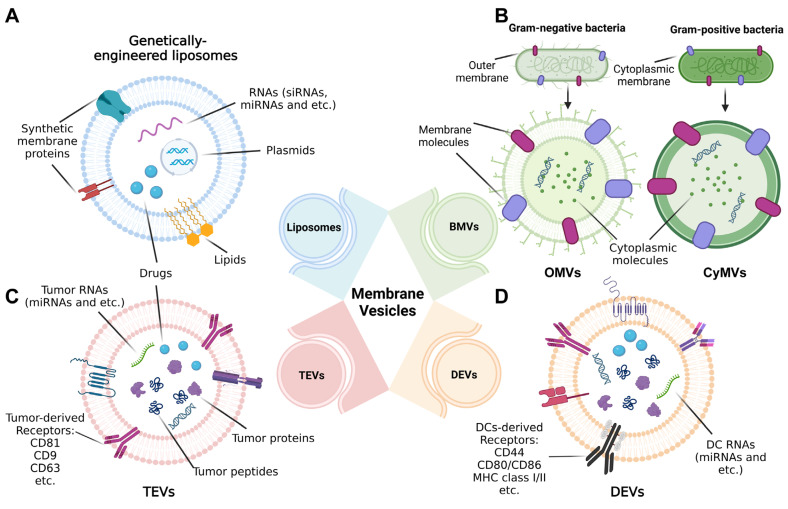
Structure of nanoscale vesicles standing for tumor vaccines in the range of biomembrane-based nanovesicles. (**A**) Liposomes are genetically engineered with genes (including RNAs and plasmids) and proteins or modified with drugs or other small molecules. (**B**) For bacteria membrane vesicles (BMVs), outer membrane vesicles (OMVs) and cytoplasmic membrane vesicles (CyMVs) are, respectively, representatives of Gram-negative and Gram-positive bacteria-generated membrane vesicles, whose structures separately rely on the origins of bacteria. (**C**) Extracellular vesicles (EVs) derived from tumor cells (TEVs) and (**D**) Evs secreted by dendritic cells (DCs); namely, DEVs are typical cell-membrane-based vesicles widely used for tumor immunotherapy, and their composition depending on parental cells presents different forms, for example, types of membrane receptors.

**Figure 2 pharmaceutics-14-02446-f002:**
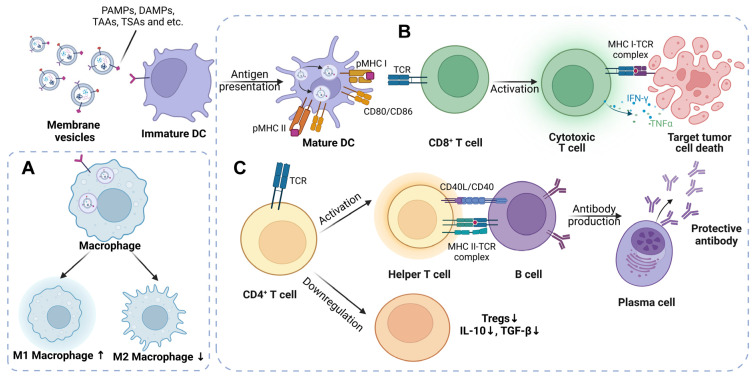
Role of membrane vesicles as tumor vaccines in immune modulation. Membrane vesicles carrying stimulators (such as tumor antigens/peptides, adjuvants, RNAs, etc.) have potentials to arouse antitumor innate and acquired immune responses. (**A**) The modified vesicles interact with macrophages and induce polarization of M2 (a protumor type) to M1 (an antitumor type) phenotype. (**B**) The vesicles containing specific antigens or peptides can be presented by DCs to further stimulate CD8^+^ T cells, causing cytotoxic T-lymphocyte (CTL) antitumor response. (**C**) In addition, during the process of antigen presentation, CD4^+^ T cells could be activated as well, leading to long-term memory immunity and downregulation of Tregs, a predominant type of immunosuppressive T cells in tumor microenvironment. ↑ means upregulation, while ↓ stands for downregulation.

**Table 1 pharmaceutics-14-02446-t001:** Fundamental applications of liposomes as tumor vaccines.

Modification Strategy	Targeting Tumor Types	Mechanisms and Outcomes	Year, Reference
Cationic liposome encapsulated with mRNA encoding cytokeratin 19	Lung caner	DC maturation (CD86 ↑; MHCII ↑); cytokine elevation (IL-12 ↑, TNF-α ↑, IL-2 ↑, IL-4 ↑). Induction of an antitumor immune response.	2020, [87]
Liposomes enveloped with ErbB-2 and OVA peptide	Lung carcinoma cells; breast cancer	ErbB-2 (known as Her-2) activates B cells to generate antibodies targeted by Pertuzumab; OVA provides T cell support.	2020, [88]
Liposomes carrying tumor antigens Gangliosides	Pancreatic cancer	Ganglioside liposomes bind to CD169 and are internalized by CD169^+^ DCs and macrophages causing cytokine production, robust cross-presentation and specific activation of CD8^+^ T cells.	2020, [89]
pH-sensitive liposomes containing OVA and α-GalCer	T lymphoma	Induction of OVA-specific IgG1 and IgG2b antibody responses; increased production of IFN-γ and IL-4; prophylactic vaccination efficacy.	2018, [90]
Liposomes containing HPV16 E7 peptide and CpG oligodeoxynucleotides and modified with DC-targeting mannose	Cervical cancer	Increased proportions of CD4^+^ and CD8^+^ T cells and CTL; reducing numbers of inhibitory immune cells such as MDSCs.	2020, [91]
Liposomes conjugated with adjuvant cRGD	Lung cancer; melanoma; breast cancer; liver cancer	cRGD promotes immunogenic cell death; cRGD-liposomes increase cellular accumulation of thymidine conjugate and enhance cytotoxicity following UVA activation.	2019, [92]
Liposomes admixed with HPV-16 E7 epitope	Cervical cancer	Induction of antigen-specific CD8^+^ T cells and production of relevant cytokines (TNF-α and IL-2); increased percentages of central and effector memory T cells.	2021, [93]
Liposomes modified with the adjuvant system including CoPoP, PHAD and immunostimulatory molecules QS-21	Colon cancer	CoPoP induces particle formation of peptides; particle-based peptides are better taken up by APCs and are represented on an MHC-I surface; generation of antigen-specific CD8^+^ T cells.	2021, [94]
Liposomes modified with ICG and pardaxin peptide	Melanoma	Under NIR, the liposomes induce the release of DAMPs and TAAs with high immunogenicity.	2022, [95]
RNA-loaded magnetic liposomes	Glioblastoma	Iron oxide enhances DCs transfection and enables tracking of DCs migration with MRI, thus predicting individual treatment effects	2019, [96]
Liposomal nanoparticles composed of mRNA (containing Ψ and 5meC) and α-GalCer	Melanoma; lymphoma	The nanosystem leads to the activation of iNKT after presented by APCs, and then cytokines (IFN-γ, IL-4, etc.) secreted by iNKT activate DCs and CTL.	2019, [97]
Liposome-decorated cancer cell membrane enveloping a plasmid encoding shRNA against Pvt1	Colorectal cancer	The biolipid nanoparticles strengthen Oxa-induced ICD; activation of DCs; inhibition of MDSCs; generation of immune memory responses for tumor ectopic rechallenging and metastasis.	2022, [98]
Cationic liposome encapsulated with tumor-derived mRNA	Melanoma; lung cancer	Increased coexpression of CD11c and PD-L1 in host-myeloid cells sensitize immunologically “cold” tumor; PD-L1^+^ APCs elicit IFN-γ production causing expansion of specific CD8^+^ T cells; combination with ICIs enhances T cell activity and synergistic antitumor efficacy.	2018, [99]
MMP2 responsive folate-modified liposome carrying doxorubicin	Breast cancer	Elimination of M2-TAMs resulting in a decrease in immunosuppressive cytokines and Treg cells, ensuring antitumor effector T cells; promotion of DCs maturation and immunostimulatory cytokines secretion.	2019, [100]

Abbreviations: CpG, cytosine–phosphate–guanine; OVA, ovalbumin; α-GalCer, α-galactosylceramide; CoPoP, cobalt–porphyrin; PHAD, monophosphoryl lipid A; ICG, indocyanine green; MRI, magnetic resonance imaging; NIR, near-infrared irradiation; Ψ, pseudouridine; 5meC, 5-methylcytidine; iNKT, invariant natural killer T cells; shRNA, short hair-pinned RNA; Pvt1, plasmacytoma variant translocation 1; ICIs, immune checkpoint inhibitors; Oxa, Oxaliplatin; ICD, immunogenic cell death; MMP2, matrix metalloprotease 2; TAMs, tumor-associated macrophages; TAAs, tumor-associated antigens; BMDCs, bone-marrow-derived DCs. ↑: Upregulation of certain subpopulations of DCs or cytokines.

**Table 3 pharmaceutics-14-02446-t003:** Fundamental applications of TEVs as tumor vaccines.

Modification Strategy	Targeting Tumor Types	Mechanisms and Outcomes	Year, Reference
Tumor-derived antigenic microparticles (T-MPs) carrying nanoFe_3_O_4_ and adjuvant CpG	Melanoma; colon cancer	Nanomaterials absorbed by APCs elicit antigen-specific host immune responses; reversion of tumor-associated macrophages into a tumor-suppressive M1 phenotype; increased infiltration of CTL.	2019, [112]
Irradiated tumor-cell-derived EVs	Hepatoma; breast cancer	Radiation endows TEVs with tumor antigens (for example, CDCP1) and HSP; enhanced infiltration of CD8^+^ and CD4^+^ T cells and activation of CTL.	2020, [113]
TEV surface modification with glycocalyx and removal of sialic acids	Glioblastoma	Increased internalization by DCs via receptor-mediated glycan-depending targeting to DCs.	2019, [114]
α-LA-engineered cancer exosomes loaded with ICD stimulators (ELANE and TLR3 agonist Hiltonol)	Breast cancer	Homing to the tumor sites and induction of ICD in cancer cells; activation of cDC1s and tumor-reactive CD8^+^ T cells.	2022, [115]
TEVs mixed with an oligonucleotide duplex and assembled with CpG-DNA	Melanoma	TEVs prolong residence in tumor tissue and activate DCs more efficiently than tumor or fibroblast cells.	2019, [116]
Exosomes derived from immunogenically dying tumor cells and modified with MART-1 and CCL22 siRNA	Pancreatic cancer	MART-1 peptide can expand T-cell-related responses; CCL22 siRNA inhibits the communication between DCs and Tregs via the CCR4/CCL22 axis.	2022, [117]
irradiated C6 (malignant glioma-cell-derived EVs	Glioblastoma	Increased percentages of apoptotic tumor cells and helper, cytotoxic and regulatory T cells.	2019, [118]
TEVs derived from irradiated cancer cells	Breast cancer	TEVs transfer dsDNA to promote production of IFN-γ via cGAS/STING pathway; TEVs evoke specific antitumor responses of CD8^+^ T cells and perform prophylactic vaccination.	2018, [119]
TEVs carrying adjuvant HMGN1	Hepatocellular carcinoma	TEVs potentiate immunogenicity and activate DCs; TEVs promote DCs homing to lymphoid tissues and augment memory lymphocytes.	2020, [120]
TEVs modified with microRNA (miR-155, miR-142 and let-7i)	Breast cancer	Induction of DCs maturation by detecting expression of MHCII, CD80 and CD40; microRNA-targeting genes (IL-6, TGFβ, IFN-γ, TLR4, SOCS1, etc.) are confirmed to mature DCs.	2019, [121]
TEVs derived from leukemia cells whose PD-L1 have been downregulated by PD-L1 shRNA	Leukemia	Modified TEVs evoke DCs maturation, T-cell activation and release of Th1 cytokine.	2022, [122]

Abbreviations: HSP, heat-shock proteins; CDCP1, CUB domain-containing protein 1; α-LA, α-lactalbumin; ELANE, neutrophil elastase; cDC1s, type one conventional DCs; HMGN1, high mobility group nucleosome-binding protein 1.

**Table 4 pharmaceutics-14-02446-t004:** Fundamental applications of DEVs as tumor vaccines.

Modification Strategy	Targeting Tumor Types	Mechanisms and Outcomes	Year, Reference
DEVs loaded with MBPN-TCyP (an AIE-photosensitizer)	Breast cancer; colon cancer	The modified DEVs induce ICD and immune-modulation function like parental DCs; DEVs synergize photodynamic immunotherapy.	2022, [124]
DEVs derived from A-Pas chiRNA-transfected DCs	Esophagus cancer	DEXs induce DC maturation (upregulation of CD83, CD86, MHC-I and MHC-II) and CD8^+^ T-cell-mediated antitumor responses.	2022, [125]
DEVs assembled with tumor peptide P47-P, AFP and immunomodulators N1ND-N	Hepatocellular carcinoma	DEVs promote DCs recruitment, activation, cross-presentation of antigens; DEVs induce antitumor responses by increasing IFN-γ^+^CD8^+^ effector T cells.	2022, [81]
Exosomes derived from AFP-expressing DCs	Hepatocellular carcinoma	DEVs remodel TME by increasing IFN-γ^+^CD8^+^ T cells and cytokines (IFN-γ and IL-2) and by decreasing CD25^+^Foxp3^+^ Treg and cytokines (IL-2 and TGF-β).	2017, [126]
DEVs conjugated with MUC1 glycopeptide antigen	Melanoma	Induction of MUC1-specific IgG antibody; activation of CTL against MUC1-positive tumor cells.	2022, [127]
DEVs derived from OVA-pulsed and activated dendritic cells modified with antiCTLA-4 antibody	Melanoma	DEVs target to T cells and activate tumor-specific T-cell responses; CTLA-4 in DEVs block inhibitory immunity and enhance the specific responses by T cells.	2020, [128]
DEVs loaded with antigen E749-57 peptide and inducer poly(I:C)	Cervical cancer	Activation of CTL; Promoted immunity of vaccinated mice splenocytes.	2018, [129]
DEVs derived from tumor cell lysate-pulsed DCs	Lung cancer	Induced proliferation of allogeneic T cell, including the subpopulation of CD3^+^Vγ9 T and CD8^+^ T cells; Activated cytotoxicity of alloPBMCs against tumor cells.	2020, [130]

Abbreviations: AIE, aggregation-induced emission; A-Pas chiRNA, a cancer-specific aberrant transcription-induced chimeric RNA; AFP, α-fetoprotein; N1ND-N, nucleosome-binding protein 1; TME, tumor microenvironment; Treg, regulatory T cells; TGF-β, transforming growth factor-β; alloPBMCs, allogeneic peripheral blood mononuclear cells.

**Table 5 pharmaceutics-14-02446-t005:** Representative clinical trials applying tumor vaccines.

Trial ID	Phase	Status	Intervention	Applied Conditions
NCT04163094	Phase 1	Active, not recruiting	A liposome-based mRNA vaccine combined with chemotherapy.	Ovarian cancer
NCT01915524	Phase 1	Terminated	RNActive^®^-derived cancer vaccine coding for tumor antigens.	Nonsmall cell lung carcinoma
NCT01052142	Phase 1	Completed	A liposomal vaccine.	Melanoma
NCT01095848	Phase 1	Completed	DPX-0907 consists of seven tumor-specific HLA-A2-restricted peptides, a universal T-Helper peptide, a polynucleotide adjuvant, a liposome, etc.	Ovarian, breast and prostatic neoplasms
NCT00623831	Phase 1	Completed; has Results	Mixed bacterial vaccine.	Melanoma, sarcoma, gastrointestinal stromal tumor, etc.
NCT02010203	Phase 1/2	Terminated; has Results	HS-410: a vaccine derived from irradiated cancer cells genetically engineered to continually secrete gp96; BCG: a vaccine derived from a live bacterium.	Bladder cancer
NCT03762291	Phase 1	Recruiting	CVD908ssb-TXSV: Salmonella-based survivin vaccine.	Multiple myeloma
NCT02657460	Phase 2	Unknown	Tumor-derived microparticles packaging chemotherapy drugs.	Malignant pleural effusion
NCT01550523	Phase 1	Completed	Exosomes from autologous glioma cells combined with an antisense molecule.	Malignant glioma of brain
NCT00020462	Phase 1	Completed	Autologous tumor cell vaccine plus interleukin-2.	Lymphoma
NCT05559177	Early Phase 1	Recruiting	Personalized chimeric exosome tumor vaccines.	Recurrent or metastatic bladder cancer
NCT00065442	Phase 3	Completed; has Results	Sipuleucel-T: Autologous antigen presenting cells loading with PA2024.	Prostate Cancer
NCT02693236	Phase1/2	Unknown	Monocyte-derived dendritic cells (moDCs) combined with cytokine-induced killer cells.	Squamous cell carcinoma of esophagus
NCT01159288	Phase 2	Completed	Dex2: tumor antigen-loaded dendritic-cell-derived exosomes.	Nonsmall cell lung carcinoma

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
