# Peer review of "Phospholipid-Membrane-Based Nanovesicles Acting as Vaccines for Tumor Immunotherapy: Classification, Mechanisms and Applications"

_pharmaceutics, 2022, doi:10.3390/pharmaceutics14112446_

Round 1

Reviewer 1 Report

The review is well written and very comprehensive regarding the subject proposed. I would suggest to better clarify the figures by adding A, B... and describe each panel below. Some recent publications on the subject could having been added in addition to potential side effects of nanovesicles acting as vaccines.  Some grammar and spelling mistakes were found, please check.

Best wishes

Author Response

Response to Reviewer 1 Comments

Point 1: I would suggest to better clarify the figures by adding A, B... and describe each panel below.

Response 1: Thanks for your good suggestion. We have added A, B and other letters to the Figure 1 and 2, and the related legends have been renewed (line 211-219 and line 419-425).

Point 2: Some recent publications on the subject could having been added in addition to potential side effects of nanovesicles acting as vaccines. 

Response 2: Thanks for your constructive suggestion. To complete the content about the safety or challenges of membrane nanovesicles and make it more convincing, we have added some recent findings in section 5 (line 556-559, 572-575, 583-585) and section 6 (line 646-653, 657-665), and these latest publications are cited simultaneously.

Point 3: Some grammar and spelling mistakes were found, please check.

Response 3: Thanks for your good suggestion. We have improved our description. At the same time, we corrected spelling mistakes and grammatical errors occurring in our manuscript (line 34, 62, 64, 117, 148, 173, 312, 404, etc.).

Reviewer 2 Report

Generally speaking, the manuscript topic is quite interesting particularly if it is intended to be published in a special issue. Bacterial-based vesicles topics can be assumed as the one the most up-to-date strategies in this field that make this review literally a well-timed one.

I have some comments that I think worth considering:

1- Since CMV is commonly known as cytomegalovirus, I suggest to use another abbreviated form on behalf of cytoplasmic membrane vesicles such as CyMV or CMeV etc.

2- In figure 1, TEVs and DEVs full forms are not mentioned in figure legend. Kindly add the full form.

3- Clinical applications are mentioned in section 5 including clinical trials. Due to significance of trail and for possibility of quick pursuing the case for the readers those who are in time-shortage, mainly clinicians, I suggest arranging the done and ongoing clinical trials in a separate table.  

Author Response

Response to Reviewer 2 Comments

Point 1: Since CMV is commonly known as cytomegalovirus, I suggest to use another abbreviated form on behalf of cytoplasmic membrane vesicles such as CyMV or CMeV etc.

Response 1: Thanks for your good suggestion. We have replaced the original confusing abbreviation “CMVs” with “CyMVs” on behalf of cytoplasmic membrane vesicles (line 142 and 214, and Figure1).

Point 2: In figure 1, TEVs and DEVs full forms are not mentioned in figure legend. Kindly add the full form.

Response 2: Thanks for your constructive suggestion. The full names of TEVs and DEVs have been given in Figure 1 legend (line 216 and 217).

Point 3: Clinical applications are mentioned in section 5 including clinical trials. Due to significance of trail and for possibility of quick pursuing the case for the readers those who are in time-shortage, mainly clinicians, I suggest arranging the done and ongoing clinical trials in a separate table.

Response 3: We are appreciated for your constructive suggestion. To make the description of clinical trials in section 5 much clearer, as seen in Table 5, the important information about these trials has been displayed in the separate table (line 550-551, line 622).

Reviewer 3 Report

This is a very well written article. The authors made an exceptional job to be extensive and analytic. However, a reference to the epigenetics of cancer should be made. This article provides every necessary information and can be cited Gougousis S, Petanidis S, Poutoglidis A, Tsetsos N, Vrochidis P, Skoumpas I, Argyriou N, Katopodi T, Domvri K. Epigenetic editing and tumor-dependent immunosuppressive signaling in head and neck malignancies. Oncol Lett. 2022 Jun;23(6):196

Author Response

Response to Reviewer 3 Comments

Point 1: This is a very well written article. The authors made an exceptional job to be extensive and analytic. However, a reference to the epigenetics of cancer should be made. This article provides every necessary information and can be cited Gougousis S, Petanidis S, Poutoglidis A, Tsetsos N, Vrochidis P, Skoumpas I, Argyriou N, Katopodi T, Domvri K. Epigenetic editing and tumor-dependent immunosuppressive signaling in head and neck malignancies. Oncol Lett. 2022 Jun;23(6):196

Response 1: We are delight for your positive and constructive comments and suggestions on our manuscript. After reading the reference to the epigenetics of cancer and relevant regulation of tumor immune microenvironment, we cited the meaningful review article and then enriched the contents in the mechanism section, to indicate the vital role of epigenetics in modifying membrane nanovesicles into tumor vaccines (line 363-372).
